# Structural Insights into Ligand—Receptor Interactions Involved in Biased Agonism of G-Protein Coupled Receptors

**DOI:** 10.3390/molecules26040851

**Published:** 2021-02-06

**Authors:** Krzysztof Jóźwiak, Anita Płazińska

**Affiliations:** Department of Biopharmacy, Medical University of Lublin, ul. Chodźki 4A., 20-093 Lublin, Poland; anita.plazinska@umlub.pl

**Keywords:** ligand directed signaling, β-adrenergic receptors, dopaminergic receptors, muscarinic receptors, serotonin receptors, opioid receptors, angiotensin receptors

## Abstract

G protein-coupled receptors (GPCRs) are versatile signaling proteins that mediate complex cellular responses to hormones and neurotransmitters. Ligand directed signaling is observed when agonists, upon binding to the same receptor, trigger significantly different configuration of intracellular events. The current work reviews the structurally defined ligand – receptor interactions that can be related to specific molecular mechanisms of ligand directed signaling across different receptors belonging to class A of GPCRs. Recent advances in GPCR structural biology allow for mapping receptors’ binding sites with residues particularly important in recognition of ligands’ structural features that are responsible for biased signaling. Various studies show particular role of specific residues lining the extended ligand binding domains, biased agonists may alternatively affect their interhelical interactions and flexibility what can be translated into intracellular loop rearrangements. Studies on opioid and angiotensin receptors indicate importance of residues located deeper within the binding cavity and direct interactions with receptor residues linking the ortosteric ligand binding site with the intracellular transducer binding domain. Collection of results across different receptors may suggest elements of common molecular mechanisms which are responsible for passing alternative signals from biased agonists.

## 1. Introduction

G-Protein coupled receptors (GPCRs) comprise a family of membrane proteins which transmits an extracellular signal into the cell interior by coupling to intracellular G proteins what eventually triggers downstream secondary cascades [1,2]. Although GPCRs all share characteristic fold of seven transmembrane helices (TM) and correlated molecular mechanisms of signal transduction, they are enormously versatile; unique structural features of an individual GPCR allow precise recognition of defined extracellular stimulus (chemical or physical) and initiate highly specific biochemical response at the intracellular level [3,4]. Functional investigations in recent years unravel a complex nature of signaling phenomenon, extracellular ligand modulation usually leads its receptor to activate a series of distinct signaling events in the cell [4,5,6]. In addition to canonical pathways regulated by coupling to a G protein, other G-protein independent signaling can be registered usually associated with arrestin recruitment or direct interactions with cellular kinases [7,8]. The current status of GPCR studies envisions the receptor macromolecule as a shapeshifting system which navigates between conformational transitions in response to interactions with either extracellular ligand(s) or intracellular binding partner(s) [9]. Ligand directed signaling is a phenomenon observed when different agonists upon binding to the same receptor trigger significantly altered pattern of interactions with intracellular transducers (G-proteins and/or arrestin), presumably inducing distinct distribution of active conformations of the receptor [10]. Therefore, structurally altered molecules, nominally agonists of a given receptor, can produce qualitatively different cellular response. The phenomenon is also termed biased agonism and its important consequences in current functional, pharmacologic and medicinal chemistry studies of various GPCRs are reviewed elsewhere [11,12].

Class A (rhodopsin-like family) is the largest cohort of GPCRs, which includes receptors sensing light, many hormones, neurotransmitters and other endogenous and exogenous stimuli [13]. Members of the family are important drug targets, thus extensively characterized by medicinal chemists and pharmacologists, also from the ligand directed signaling standpoint. Structural biology studies are resolving molecular mechanisms of activation in many representatives of this class of GPCRs as well as their mechanisms of ligand recognition [14]. The goal of the current work is to review the structurally defined ligand – receptor interactions that can be specifically related to molecular mechanisms of ligand directed signaling across different receptors. While we do not intend to provide a comprehensive review of biased agonism phenomenon and the readers are referred to specialized articles describing ligand bias in specific receptors, the work is focused on interactions between ligands’ structural features and protein residues that can be associated with biasing the signaling in a given receptor. Therefore, specific intermolecular interactions within agonist binding site can be postulated to play a role in aiming the receptor to signal into the agonist specific direction. Class A receptors are known to share similar molecular rearrangements during activation transition that link the ligand binding domain and the intracellular interface binding to a transducer [13]. Inspection of available ligand – receptor interactions involved in biased agonism across various receptors also suggests the elements of common mechanisms in recognition and receptor response to biased ligands that are shared across different receptors. The work describes the following receptor systems which are well characterized by structural biology and their ligand – receptor interactions can be linked with differences in signaling pattern upon activation: β-adrenergic receptors, dopaminergic D2L receptor, muscarinic M2 receptor, serotonin 5HT_2B_ receptor, opioid receptors and angiotensin AT_1_ receptor.

## 2. β-Adrenergic Receptors

β-adrenergic receptors (β-ARs) are the most structurally and pharmacologically studied subgroup of GPCRs [15,16]. They are also important drug targets, where both beta- mimetics and beta-blockers are employed in therapies of various conditions [17,18,19]. One of the first structurally evident instances of biased agonism at the β-AR system was characterized for bucindolol, carvedilol and nebivolol. These compounds are known beta blockers and the latter two are approved drugs for treatment of hypertension or congestive heart failure [20]. Their main function is to attenuate the receptor’s mediated G_s_ signaling, however, in contrast to other known compounds of similar pharmacological function, their binding to the β-ARs was shown to induce additional cellular response regulated by non-G-protein pathways. Warne et al. reported an X-ray crystallography study, where a thermostabilized turkey variant of the β_1_-AR was complexed with two such biased ligands, bucindolol and carvedilol [21]. The two ligands share the main structural scaffold of other classical beta blockers, both, however, carry a bulky extension of the aminoalkyl tail, (1*H*-indol-3-yl)-1,1-dimethylethyl- and (2-methoxyphenoxy)ethyl]- for bucindolol and carvedilol, respectively. These structural features are considered to be responsible for their extra activity inducing G_s_ independent signaling [22]. In the crystal structures of bucindolol–β_1_-AR and carvedilol–β_1_-AR complexes ligand molecules assumed typical orientation within the receptor binding. As other beta blockers, both the bucindolol and carvedilol bound complexes did not exhibit the contraction of the binding pocket characteristic for the structures with full and partial agonist bound, an indication that these two ligands did not induce the initial conformational changes in the receptor needed to activate G proteins [21]. Their bulky aromatic moieties at the aminoalkyl end occupied the so-called extended ligand binding domain (ELBD), the space between the upper part of TM6 and TM7 capped by the extracellular loop 2 (ECL2). The work postulated that additional contact between that part of the ligand and ELBD domain could lead to increased probability of subtle conformational changes that might be transmitted to the receptor’s C-terminus where phosphorylation by GPCR-specific kinases promotes binding of arrestin [21]. In a consecutive study from the same group, ligand interactions with the ELBD were further characterized with thermostabilized β_1_-AR receptor co-crystalized to a conformation specific nanobody [23]. Therefore, the previously obtained structures representing an inactive state of β_1_-AR could be compared with a nanobody stabilized active conformations. At the ligand binding domain, a significant decrease in the volume of the cavity was characterized as a main effect of transition from inactive into active conformation. That was mainly due to inward movements of the extracellular ends of TM5, TM6 and TM7, however, the authors identified reorientation of certain residues forming the ELBD as a critical change during activation. Specifically, the pincer-like movement of the two residues, F325^7.35^ (in the whole text, the superscript numbers the TM residues according to the Ballesteros-Weinstein scheme [24]) and F201^ECL2^ and towards the ligand had the largest effect on reduction of the volume of the binding pocket [23].

While the canonical part of the binding pocket in β-adrenergic receptors remains highly evolutionary conserved, the residues forming ELBDs may differ depending on the receptor subtype and this fact is widely exploited by subtype specific ligands. In particular, the β_2_-AR selective ligands contain elaborated substructures at their aminoalkyl tails aimed to explore the ELBD of the β_2_-AR. Remarkably, a very peculiar example of biased agonism at the β_2_-AR subtype was reported and related to the specific structural features introduced to β_2_-AR selective agonists. Studies by Xiao et al., indicated that β_2_-AR upon agonist activation couples to G_s_ and, to a lesser extent, G_i_ proteins [25]. The two events have the opposite effects on studied cell system (rat cardiomyocytes) affecting positively (G_s_-coupling) and negatively (G_i_-coupling) contractility of the cells [26]. The G_i_ coupling can be blocked by pertussis toxin (PTx) and adding PTx to the experiment resulted in significantly increased potency of many β_2_-AR agonists e.g., salbutamol, procaterol, zinterol, etc. [25]. Fenoterol, a selective β_2_-AR agonist was characterized as an exemption, activation pattern induced by that ligand in cardiomyocytes appeared to be PTx independent. The observation was interpreted as biased agonism, fenoterol and later some of its derivatives activated the receptor to a form that coupled G_s_ protein only, while other typical agonists induce dual, [G_s_ + G_i_] coupling. [27]. Our group performed a medicinal chemistry project by chiral switching and aminoalkyl tail optimization of fenoterol scaffold aimed at development of highly selective β_2_-AR agonists [28,29,30]. The project generated a number of structures with diverse biological and pharmacological activities [31,32,33,34]. This congeneric cohort of compounds proved to be very effective chemical biology tool in structural characterization of β_2_-AR activation patterns leading to either G_s_ selective coupling (PTx insensitive) or dual, [G_s_ + G_i_] coupling (PTx sensitive). Table 1 shows a selection of structures generated in the project along with their potency to activate the receptor to either G_s_ exclusive or dual signaling pattern [27]. While many tested molecules induced [G_s_ + G_i_] profile, three of the derivatives (*R*,*R*)-01, (*R*,*R*)-02 and (*R*,*R*)-03 activated β_2_-AR at cardiomyocytes in PTx insensitive manner. SAR analysis allowed assigning key structural condition at the aminoalkyl tail of those three compounds, a hydrogen bond acceptor atom attached at the 4′ position of benzene ring system. It led to the hypothesis that the presence of –OH, –OCH_3_ or –NH_2_ substituent at that position was essential to form a specific interaction with the receptor, which might be a key event to induce specific activation of the β_2_-AR leading to G_s_ selective coupling. Subsequent docking simulations postulated that such an interaction may be realized by hydrogen bond (HB) created between a 4′-substituent of a ligand and hydroxyl group of Y308^7.35^ within the ELBD. Molecular modeling observation was further confirmed by mutagenesis studies, Y308^7.35^A mutation introduced to a receptor resulted in significant drop of the affinity of the three G_s_ selective ligands in comparison of the affinity on the wild type. When other derivatives, inducing dual, [G_s_+G_i_] signaling pattern were tested (e.g., (*R*,*R*)-05, (*R*,*R*)-06 and (*R*,*R*)-07, Table 1), their affinities to the Y308^7.35^A mutant remained at the level measured previously at the wild type receptor. Another set of experiments on additional mutant, Y308^7.35^F evidenced that such mutated receptor regained the PTx sensitive pattern of activity when activated by (*R*,*R*)-01, (*R*,*R*)-02 and (*R*,*R*)-03 [27]. Taking together, all above results strongly suggested that unusual behavior of (*R*,*R*)-01, (*R*,*R*)-02 or (*R*,*R*)-03 eliciting exclusively G_s_ cellular response could result from capability of these ligands to form a HB with Y308^7.35^ residue located at ELBD. Figure 1A shows postulated mechanism of molecular interactions between (*R*,*R*)-02, a G_s_ signaling ligand forming HB with Y308^7.35^ residue. The complex is compared with results of docking simulations of (*R*,*R*)-05, a ligand lacking hydrogen-bond acceptor at 4′-position and inducing dual [G_s_ + G_i_] signaling pattern, Figure 1B, where Y308^7.35^ residue forms alternative interhelical HB with nearby N293^6.55^ residue.

## 3. Dopaminergic Receptors

Growing body of evidence indicates that ELBD is an important structural hotspot to elicit ligand directed signaling patterns in other GPCRs as well. Tschammer et al. in their work on 1,4-disubstituted aromatic piperidines/piperazines acting on dopaminergic D2L receptor identified that H393^6.55^ played a crucial role in imposing a ligand directed signaling in the system [35]. The operational model of agonism was used to quantify the ligand bias between the ability of the compounds to inhibit cAMP accumulation or stimulate intracellular ERK1/2 phosphorylation and substantial ligand biased signaling was observed for the wild type receptor. When the H393^6.55^A variant of the receptor was tested, the overall increase of agonism was observed, however, the system lost its ability to produce biased signaling imposed by the agonists [35]. In another work, Fowler et al., studied D2L signaling in various cellular responses; adenylate cyclase, MAPK, arachidonic acid release, and guanosine 5′-*O*-(3-thio)triphosphate binding produced by three rigid agonists, dihydrexidine, dinapsoline and dinoxyline [36]. Alanine substitution approach was employed to explain the role of three serine residues located on TM5, S193^5.42^, S194^5.43^ and S197^5.46^ and the functional studies of the mutants showed that S193^5.42^A and S197^5.46^A mutations abolished the activity of dopamine and three other ligands although dihydrexidine retained intrinsic activity at MAPK function only with S193^5.42^A. Remarkably, S194^5.43^A mutation did not affect the intrinsic activity for adenyl cyclase and MAPK for any of the ligands, but eliminated arachidonic acid release activity for dinapsoline and dihydrexidine but not dinoxyline [36]. That was interesting observation particularly from the structural point of view, TM5 located S194^5.43^ residue faced the direction of TM6 and was capable to form a HB interaction with H393^6.55^ residue. Parallel molecular modeling studies by the same group postulated a possibility of two alternative ELBD conformations which could be responsible for different recognition of agonists and subsequently biased signaling of downstream receptor effects. The central role was played by H393^6.55^ residue swinging between two positions; in the first state the residue was shifted towards TM5 and formed interhelical HB with S194^5.43^ residue, whereas in the second state the H393^6.55^ residue was placed more towards TM7 and created alternative HB bridge with Y429^7.35^ residue [36]. Interestingly, the two-conformation mechanism proposed to control alternative ligand directed signaling patterns in D2L receptor highly resembles the mechanism proposed by Woo et al., for explanation of G_s_ vs. [G_s_+G_i_] alternative signaling observed in β_2_-AR system. Both mechanisms are realized by the residues being positional equivalents in the respective ELBDs. In addition, in both cases two alternative interhelical HBs has been postulated, the one linking residues at positions ^7.35^ and ^6.55^, the other connecting residue ^6.55^ with residue ^5.43^, see Figure 1.

Other elements of structural requirements for G protein-biased agonist activity in the D2L system were described in studies which elucidated four structural features that were crucial for agonist efficacy and signaling bias for MLS1547 and its derivatives [37,38,39]. One of the most important determinants for G protein-biased signaling was the interaction of quinoline ring of the ligand molecule with a hydrophobic pocket comprised of I184^ECL2^, F189^5.38^, and V190^5.41^ residues. The two HBs created between: (i) nitrogen atom in the *ortho*-position of the pyridine moiety of MLS1547 and T412^6.54^ of D2L and (ii) hydroxyl group at the quinoline ring of MLS1547 and D114^3.32^ were found crucial for the G biased signaling. Additionally, an important HB involving a conserved aspartate residue D114^3.32^ can be formed by the positively charged nitrogen atom at the piperazine moiety of MLS1547 [37]. Moreover, the chlorine atom at the quinoline ring of MLS1547 was hypothesized to interact with hydrophobic pocket and prevent the tilting of TM5 during receptor activation. The structural analogues of MLS1547 that dispossessed the chlorine atom at the quinoline ring presented a modest decrease in agonist potency and loss of signaling bias [38]. Additionally, F189^5.38^ was identified as a micro-switch that regulates the active state of D2L for recruiting β-arrestin [39]. Interestingly, such a switch exists not only in dopamine receptor but also in several other related GPCRs, including the β_2_-AR [40] and 5-HT_2B_ serotonin receptors [41].

## 4. Muscarinic Receptors

Muscarinic acetylcholine receptors are another member of class A GPCRs, where ELBD is extensively exploited by medicinal chemists, particularly in development of allosteric molecules exploring this site [42]. Studies showed that allosteric interaction within ELBD altered the receptor signaling pattern and could be used as a fine tuner of downstream signaling. For example, Bock et al. developed a cohort of hybrid compounds targeting M2 receptor in a dualsteric manner: the iperoxo building block while targeting the ortosteric site with super high affinity was linked to phtalimide or naphtalimide buinding blocks designated to allosterically interact with ELBD; hexamethylene or octamethylene spacer was used to link the two blocks [43]. The authors characterized synthesized structures by the bias plot comparing M2 receptor signaling via G_i_ ([^35^S]guanosine -5′-*O*-(3-thio)triphosphate binding to membranes of CHO-hM_2_ cells) and G_s_ (cAMP accumulation in CHO-hM_2_ cells pretreated with PTx) routes. The study showed that, while acetylcholine and iperoxo itself induced dual G_i_ and G_s_ signaling upon M2 binding, the hybrid molecules induced signaling where G_i_ component was much more dominant and G_s_ component significantly diminished in comparison to the control compounds. Therefore, a significant structure dependent G_i_ bias could be assigned to those dual steric probes. Molecular modeling simulations suggested that observed difference in binding modes was due to interactions of an allosteric building block with two key residues located at the ELBD, W422^7.35^ and Y177^ECL2^. The authors hypothesized that dual steric ligands induced reduction in flexibility of that domain, what can be translated into restricted intracellular loop rearrangement and subsequently to impaired downstream signaling events. The hypothesis was further explored in mutational study where W422^7.35^A alteration was introduced to the receptor. For all studied structures binding affinities were significantly reduced in the mutant when compared to the wild type receptor. But most importantly, W422^7.35^A mutant gained in efficacy towards G_s_ pathway upon binding to the studied dual-steric ligand, in comparison to the wild type [43].

The importance of W^7.35^ residue in ligand directing signaling of muscarinic receptors was further confirmed in the structural biology study of agonist bound, active state of the human M2 receptor stabilized by a G-protein mimetic camelid antibody fragment [44]. In that work Kruse et al. resolved two ligand – receptor complexes, the first contained iperoxo molecule present in the ortosteric site; the second enclosed two ligand molecules, the orthosteric iperoxo and positive modulator molecule, LY2119620 occupying the allosteric vestibule. Figure 2 shows the comparison of these two complexes focusing on differences within the ELBD. Receptor arrangements are almost identical at the ortosteric domain recognizing iperoxo molecule. Comparison at the level of allosteric site also shows high similarities, but W422^7.35^ is a noteworthy exemption; its sidechain assumed two alternative conformations depending on whether LY2119620 molecule was present at ELBD or not. In the iperoxo–M2 complex, the indole ring of the residue adopts a fairly perpendicular orientation in relation to TM7 axis (nearly parallel in relation to the membrane surface). Such conformation allows W422^7.35^ residue for flexible exploration of the cavity devoid of allosteric ligand (Figure 2A). However, such bulky residue switched that conformation while in the complex containing both iperoxo and LY2119620. As shown in Figure 2B, the indole ring of W422^7.35^ is sterically required to vacate the space for the allosteric ligand and reorients parallel to the LY2119620 ring system, parallel to TM7 axis and perpendicular to membrane surface. Such conformation has a significant consequence; the W422^7.35^ conformer while in (iperoxo + LY2119620)–M2 complex is able to create interhelical HB with N410^6.58^.

## 5. Serotonin Receptors

Lisergic acid diethylamide (LSD) is the prototypical hallucinogen that primarily acts via serotonin (5-HT) receptors. Wacker et al. resolved crystal structure of human 5-HT_2B_ receptor in complex with LSD [45] and the study revealed a conformational reorganization of the receptor ECL2 in order to accommodate that particular ligand. Subsequent molecular dynamics (MD) simulations postulated that rearrangement to be responsible for exceptionally slow dissociation rate in LSD–5-HT_2B_ and LSD–5-HT_2A_ complexes [45]. Particular role in that lid – like function of ECL2 section was assigned to L209^ECL2^ residue of the 5-HT_2B_ receptor; the side chain of that residue was directed towards LSD and acted as a bulky steric blocker fixing the ligand molecule within contracted binding domain. The authors validated their postulate by 5-HT_2B_ mutational study; the receptor where L209^ECL2^ was substituted with less bulky alanine side chain showed markedly faster off-rate of LSD when compared to the wild type readouts [38]. The study provided additional insight into structural features of the LSD–5-HT_2B_ complex that might be related to biased agonism properties. Single-point mutation of L209^ECL2^A changed not only the ligand dissociation kinetics; it additionally changed the signaling pattern; LSD upon binding to wild type 5-HT_2B_ induced signaling both via G_q_ protein coupling and via β-arrestin recruitment. In L209^ECL2^A mutant G_q_ coupling remained at relatively the same level in comparison to the wild type but the time dependent augmentation of β-arrestin recruitment was selectively attenuated [45].

5-HT receptors are known to be very sensitive in functional recognition of ligands within the binding site; even a very minute structural change of LSD molecule may translate its agonistic profile into an antagonist upon binding to the receptor [46]. In the structural study by McCorvy et al. the 5-HT_2B_ binding modes of several chemically congeneric agonists and antagonists were compared [47]. The authors attempted to identify key residues responsible for the receptor activation and to postulate differential mechanisms of recognition of receptor agonists as opposed to antagonists; additional structure – guided mutagenesis experiments revealed residues that were essential for agonist mediated biased signaling [47]. The study characterized a series of residues essential for ligand recognition at the binding site but that was L362^7.35^ residue which was associated to quantitative differential in alternative signaling patterns. L362^7.35^F mutation at 5-HT_2B_ receptor was shown to restore the agonistic properties of a nominal antagonist molecule, lisuride. However, the mutation reestablished only the G_q_ coupling signaling, while the ligand did not elicit β-arrestin-2 related agonism. Similarly, the L362^7.35^F mutation did not affect LSD’s G_q_ agonism, but it abolished β-arrestin recruitment observed for LSD on the wild type receptor [40]. Further structural studies of receptor cocrystalized with non-ergoline, purportedly 5-HT_2B_ selective antagonist, LY266097 revealed that extracellular tip of TM7 acted as a trigger for biased agonism. While the tetrahydro-β-carboline core of the ligand molecule was placed within the ortosteric site and interacted with residues Asp135^3.32^, Phe340^6.51^ and 341^6.52^, its 2-chloro-3,4-dimethoxybenzyl substituent was oriented much closer to TM7 than lisuride molecule. Functional studies determined that LY266097 was a modest G_q_ partial agonists without detectable β-arrestin-2 activity. The authors hypothesized that the extend of G_q_ agonism was determined by a ligand contact with L362^7.35^ residue. The hypothesis was elegantly verified by functional studies at the L362^7.35^F mutant, where LY266097 elicited diminished G_q_ activity suggesting that bulkier phenylalanine at that position sterically clashed with 2-chloro-3,4-dimethoxy- benzyl moiety and abolished the agonism [47].

Taken together, two residues located at the ELBD of 5-HT_2B_ receptor, Leu209^ECL2^ and Leu362^7.35^ can be directly linked with biased signaling induced by different ligands (see Figure 3). Interestingly, each of these residues can be precisely associated with two alternative signaling patterns; Leu209^ECL2^ residue has been found essential for massive β-arrestin-2 bias of LSD molecule [45], while Leu362^7.35^ emerged as involved in biasing the signaling towards G_q_ related pathways upon binding to LSD and some related molecules [47]. Notably, the two residues 5-HT_2B_ receptor are the positional equivalents of the residues involved in pincer like movement responsible for recognition of biased ligands described earlier in β_1_-AR receptor system, F201^ECL2^ and F325^7.35^.

## 6. Opioid Receptors

Three principal members of opioid receptor (OR), μ, δ and κ are important targets for pain management. Multiple lines of evidence show that the analgesic effect of μ-OR is related to G_i_ intracellular signaling induced upon receptor activation, while respiratory depression, impaired intestinal motility, opioid tolerance and addiction can be linked to the pathways regulated by β-arrestin receptor interactions. [48]. It underlines the importance of medicinal chemistry efforts to develop G_i_ biased ligands of μ-OR with minimal to no β-arrestin-2 signaling as future analgesics (for review, c.f. [49]). Numerous studies were aimed at explanation of ligand – receptor interaction that might be linked to a biased signaling of the receptor. Hothersall et al. identified two μ-OR mutations, W320^7.35^A and Y328^7.43^F that changed pathway bias with different patterns between peptide agonists. Subtype selective ligand DAMGO exhibited increase in β-arrestin activity at the W320^7.35^A mutant, while Y328^7.43^F substitution completely abrogated β-arrestin signaling. Endomorphin-1 gained the efficacies at both pathways with Y328^7.43^F mutation but lost them at W320^7.35^A. For endomorphin-2, the W320^7.35^A mutation resulted in a shift from G_i_ preferred bias at the wildtype towards β-arrestin2 bias at the mutant [50]. Cheng et al. performed molecular simulations of the G_i_ protein biased activation and inactivation mechanisms of μ-OR system [51]. MD of the receptor alone or in complex with G_i_ biased agonists, TRV130 or BU72 or antagonists, β-FNA and NTx suggested two residues, W295^6.48^ and Y328^7.43^ that acted as a paired activation switch. The authors postulated the switch to be critical for receptor activation and the ligand interactions that positioned the two residues influenced the increased or decreased signaling via β-arrestin. In more recent molecular modeling study de Waal et al. employed adaptively biased MD simulations to determine the molecular mechanisms of interactions between μ-OR system and fentanyl and two its derivatives, highly potent β-arrestin biased agonists [52]. The authors proposed an activation mechanism where agonist molecules mediated β-arrestin signaling of μ-OR through a novel M153^3.36^ “microswitch” at the ortosteric site. The residue was identified to directly interact with ligands’ N–aniline ring and its conformational change affected the rotameric state of W295^6.48^ residue. The findings were further validated by design and synthesis of novel fentanyl based derivatives with confirmed complete, clinically desirable, G_i_ protein biased coupling [52].

Che at al. presented the nanobody stabilized active state structure of another subtype, the κ-OR [53]. A combination of structural biology analyses, binding and functional assays allowed identification of key residues important as molecular determinants of κ-OR ligand binding and agonist efficacy. Among them, κ-OR residues involved in conferring different patterns of biased signaling were postulated. The authors showed for example that replacing Y312^7.35^ in the κ-OR binding pocket with tryptophan residue found in the corresponding position of μ-OR, transformed IBNtxA, a balanced κ-OR agonist into a G-protein-biased ligand (thus mimicking its activity at the μ-OR subtype). That was an elegant illustration that certain residues located at corresponding positions of closely related subtypes may be responsible for eliciting differential signaling patterns between the two receptors [53].

A structural study of another subtype of opioid receptor, human δ-OR, revealed a key role of complexed sodium ion in mediating allosteric control of receptor functional selectivity towards a specific transducer [54]. Functional studies on the δ-OR variants, where key residues forming the allosteric locus for Na^+^ ion were mutated, identified that N131^3.35^A/V augmented constitutive β-arrestin-mediated signaling, while D95^2.50^A, N310^7.45^A and N314^7.49^A mutations transformed classical δ-opioid antagonists into a β-arrestin-biased agonists. Overall, the data revealed that sodium-coordinating residues acted as “efficacy switches” at a prototypic G-protein-coupled receptor. [54]. It revealed that the sodium ion regulated the activation of a GPCR through cooperation with the ligand at the orthosteric site. Subsequently, Sun at al. presented the results of computational simulation of δ-OR system that postulated W274^6.48^ residue located adjacent to the allosteric Na^+^ binding site to establish a bridge between the sodium allosteric site and the orthosteric site. During MD simulations allosteric sodium ion exploited a distinct conformation of that key residue to propagate the modulation to TM5 and TM6, which was further transmitted along the helixes and regulated their positions on the intracellular side. The models hypothesized the contrast between the allosteric effects towards the two transducing partners of the receptor, Na^+^ allosteric modulation significantly altered the β-arrestin recruitment, while it affected the G protein to much lesser extent [55].

## 7. Angiotensin Receptor

Angiotensin receptor 1 (AT_1_) is very well characterized for its ligand directed signaling properties. Binding of receptor’s endogenous ligand, the octapeptide angiotensin II (AngII) stimulates both G_q_ protein mediated and arrestin mediated signaling pathways. Functional studies of AngII showed that its single point mutations could result in either β-arrestin – biased or G_q_ – biased response in comparison to the wild type hormone [56]. For example, F8A mutation significantly altered the AT_1_ signaling pattern and such octapeptide did not activate the receptor to elicit G_q_ dependent routes while still maintained high level of β-arrestin signaling. The receptor is a major drug target, and there is a significant interest in development of arrestin – biased AT_1_ ligands for treatment of heart failure, such ligands have clinically relevant antihypertensive effects but also improve cardiac function through β-arrestin – mediated pathways [57]. Recent structural biology report [58] positioned the AngII molecule within the AT_1_ structure locating the hormone N-terminus at the extracellular entrance and the C-terminus at the base of the binding pocket. Closer inspection of the complex indicated that AngII F8 residue pointed towards the vestibule formed by the central portions of TM3, TM4 and TM5 of AT_1_, however, that C-terminal residue of the hormone was poorly resolved with markedly high B-factors. Additionally, weak or none electron density was observed for the side chains of the two AT_1_ residues located nearby F8 of AngII, L112^3.36^ and Y292^7.43^ suggesting that region of the AngII – AT1 complex to be conformationally heterogeneous [58]. In the same study, two alternative versions of AngII molecule, F8A mutant (TRV023) and F8del (TRV026) were cocrystalized with AT_1_, and the results evidenced that both L112^3.36^ and Y292^7.43^ residues were resolved better with lower B-factors while complexed with those β-arrestin biased molecules. Additionally, clear rearrangement of the two residues could be observed, TM3 was rotated on-axis moving L112^3.36^ away from position occupied by Y292^7.43^ in comparison to the wild type AngII – AT1 complex (Figure 4). Consistent with crystallographic observations, L112^3.36^A mutation of AT_1_, a residue expected to better accommodate the flexibility of AngII F8, increased receptor affinity to the wild type AngII but decreased affinity for TRV026 or TRV023. Similarly, Y292^7.43^A substitution (which was designed by the authors to mimic disordered state induced by the hormone — see Figure 4A) increased AT_1_ affinity for AngII, but had relatively minor effects on TRV026 or TRV023. The authors also noted that both mutations, despite enhancing affinities to AngII, did not induce increased efficacy or potency in activating AngII-dependent G_q_ signaling [58].

Above structural results led to the postulate that L112^3.36^ and Y292^7.43^ residues of AT_1_ played a decisive role in inducing the receptor coupling to G_q_ protein upon binding of AngII. The hypothesis was additionally validated by extensive molecular dynamics simulations of the AngII–AT1 complex [59]. The simulations suggested that the agonist bound receptor was oscillating between two conformations referred to as canonical active conformation or alternative active conformation. During that transition TM7 twisted above its proline kink leading the intracellular portion of TM7 to shift away from TM2 and TM3. In details, the movement induced N46^1.50^ to switch hydrogen bond interaction from C296^7.47^ to N295^7.46^ and further side chains rearrangement of adjacent residues. Importantly, the authors additionally noted that those intracellular transitions of TM7 region were allosterically linked to the ligand binding region, affecting in particular Y292^7.43^ residue position. In the alternative active conformation, the residue pointed towards TM3 while in the canonical active conformation the Tyr292^7.43^ was shifted toward TM2. The change affected side chain conformations of adjacent residues in AT_1_, L112^3.36^, N111^3.35^ and F77^2.52^ as well as F8 residue of AngII. Further replica exchange MD simulations showed that β-arrestin biased ligands favored the alternative active conformation while ligands with G_q_-protein bias properties favored binding to the canonical active state [59].

## 8. Conclusions

A growing number of structural biology studies support the idea that class A GPCRs, despite having high variance in amino acid sequences, share similar three dimensional folding as well as analogous patterns of conformational transitions in response to interactions with their binding partners, intracellular transduces and extracellular ligands [10]. One of the most canonical hallmarks of GPCRs activation is the outward movement of the intracellular portions of TM5 and TM6 allowing the receptor to open the interface for interaction with a G protein [60]. Evolutionary developed variations in residues shaping this interface play a decisive role in recognition of specific type(s) of protein G, in a similar manner differences in residues lining orthosteric binding domain are critical for specific recognition of endogenous hormone or neurotransmitter. Hence, different receptors belonging to class A (having the ligand binding domain located at the bundle of transmembrane helices) maintain relatively uniform molecular mechanisms of conformational transitions upon activation, while differences in their sequences allows for enormous versatility in both recognition of agonists and signal transmission via interactions with transducing protein(s). Ligand directed signaling is a phenomenon that has been puzzling researches for years as it opens novel possibilities of using specialized ligand molecules for precise tuning of induced signaling pathway(s) via one pharmacological target. Combination of medicinal chemistry, molecular pharmacology, structural biology and mutagenesis studies sheds the light on the structural mechanisms responsible for biased agonism functionality in some representatives of class A receptors. Very interestingly, these mechanisms show significant similarities when compared across characterized receptor systems.

First group of GPCRs discussed above are receptors for neurotransmitters; small molecules like epinephrine (β-ARs) acetylcholine (M2), dopamine (D2L) and serotonin (5-HT_2B_). All these receptors can be characterized that their ortosteric sites accommodating an endogenous neurotransmitter molecule are accompanied by its extension, the ELBD. The latter domain does not seem to interact significantly with a neurotransmitter, however, many exogenous agonists including drugs introduced into clinical practice explore both the ortosteric and ELBD while binding to the receptor. Examples presented above illustrate that, while the core drug – receptor interactions occurs at the ortosteric site, additional interactions with residues lining ELBDs can play important roles in subtle tuning of observed ligand directed signaling. Noteworthy, the residues that were postulated to form essential interactions biasing the ligand functions shared analogous positioning within ELBDs of different receptors. The studies identified those hot spots residues to be typically located at upper tips of TM7 (^7.35^ position at β_1_-AR, β_2_-AR, M2, D2L and 5-HT_2B_ systems); TM6 (^6.55^ position in β_2_-AR and M2 receptor and ^6.58^ position at D2L receptor) and TM5 (^5.43^ at β_2_-AR and M2). Typically, residues at those positions were involved in interhelical HB interactions which were switched on and off while interacting with agonists of alternative signaling profiles (observed in β_2_-AR, M2 and D2L systems). Additional important position was found at the ECL2 (in β_1_-AR and 5HT_2B_ receptors), the loop acts of specific lid of many ELBDs, and the residues involved in interactions with biased agonists pointed downward the ECL2. It was suggested that ligands of alternative signaling patterns stabilized certain interactions bridging TM helices and shaped the position of ECL2, thus, affecting the flexibility of ELBD what could translate into restricted intracellular loop rearrangement and subsequently to alter downstream signaling events [36].

In contrast to neurotransmitter receptors, ORs and AT1 are GPCRs sensing endogenous peptides. These bulkier molecules natively bind to a large receptor vestibule covering completely a space previously referred to as ortosteric and ELBD domains. Studies on μ-OR and κ-OR indicated again important role of ^7.35^ residue played in recognition of agonists eliciting alternative signaling patterns. δ-OR investigations on the other hand underlined the role of region located at the bottom of the ligand binding cavity, W^6.48^ toggle switch residue was hypothesized to establish a bridge transmitting biased signals from the ortosteric site to the allosteric cavity accommodating Na^+^ ion deep in the TM helical bundle. The importance of residues lining the bottom of ortosteric site was explicitly showed in the AT_1_ system where studies of AngII molecule cocrystalized with the receptor identified conformationally heterogeneous region in the complex: F8 residue of AngII and L^3.36^ and Y^7.43^ of AT_1_. Simultaneous structural studies of β-arrestin biased AngII mutants, F8A and F8del evidenced that both residues at ^3.36^ and ^7.43^ positions showed much better resolution. Additional rearrangements of positions of these residues were observed when compared to native, wild type AngII cocrystalized structure. The observation strongly suggests that these two residues located at the center of the orthosteric site play a crucial role in inducing the receptor coupling to G_q_ protein upon binding of AngII.

While growing number of structural biology studies characterize molecular nature of specific interactions between a G-protein heterodimer and various GPCRs [6,60,61], potential mechanisms coupling the receptors to arrestins remain much more elusive. Recently, however, structures of β-arrestin-1 complexes with neurotensin-1 receptor [62] and turkey thermostable β_1_-AR [63] were reported. In the latter study, Lee et al. described cryo-electron microscopy structure of formoterol–β_1_-AR complex coupled to β-arrestin-1 in lipid nanodiscs. The authors compared their β-arrestin-1 bound β_1_-AR complex with previously reported G_s_ heterotrimer bound formoterol–β_2_-AR complex [64] and the difference in conformational arrangement of receptors was found; the finger loop of β-arrestin-1 occupied a narrower cleft on the intracellular surface and it was closer to TM7 of the receptor when compared with the C-terminal α5 helix of G_s_ [63]. The authors suggested that, upon G protein dissociation, coupling of β-arrestin-1 to the receptor was accompanied with slight inward movement of the cytoplasmic end of TM5, resulting in an outward movement of its extracellular tip; the move of the helix away from the ligand binding domain could weaken the HB interactions between formoterol and TM5 [63]. Based on that model, one can hypothesize that the β-arrestin-1 induced change in TM5 rearranges the ELBD and might affect a pattern of interactions between TM5 and adjacent helices and/or ECL2 lid. Therefore, specific parts of various agonists by fine interactions within these regions may affect the conformational state(s) that are preferential for interactions with one or another transducers eventually leading to a ligand directed signaling.

Emerging structural information on ligand – receptor interactions involved in alternative recognition of biased agonists is of the frontline importance in current and future GPCR pharmacology. Related receptor subtypes sensing the same neurotransmitter have their orthosteric sites formed by specialized and highly evolutionary conserved residues. It is not necessarily true in case of their ELBDs and the fact is widely explored by medicinal chemistry in development of ligands selective towards a specific receptor. Many such molecules effectively exercise their selectivity via “address” residues located at ELBDs that differ between subtypes. We must be aware that plenty of subtype selective ligands that have been developed so far, may express distinct, sometimes unexpected or unwelcome, signaling patterns and the fact might have clinical consequences. On the other hand, future development of subtype selective ligands opens the possibility for fine tuning of induced signaling pathways by a control of agonist – ELBD interactions. One example has been described above, fenoterol-based agonists of β_2_-AR activating the receptor to couple either G_s_ only or dually [G_s_ + G_i_] are highly selective towards that particular receptor subtype. Their cellular (and possibly physiological) effects, however, can be fine-tuned by a presence of a hydrogen bond acceptor substituent at the agonist aminoalkyl tail that controls possible ligand interaction with ^7.35^ receptor residue [27]. AT_1_, on the other hand, is a hallmark for its ligand directed signaling properties and currently approved antihypertensive drugs targeting the receptor may be considered suboptimal as they attenuate both G_q_ related signaling and β-arrestin related signaling in response to AngII [58]. The former is a desired therapeutic effect in hypertension treatment, the latter is undesired as β-arrestin signaling via AT1 is considered cardioprotective [57] and heart failure treatment may significantly benefit from development of β-arrestin biased AT_1_ agonists, such ligands can increase cardiac contractility without undesired hypertensive effects [65]. Also, novel agonists of μ-OR biased towards G_i_ signaling with minimal effect on β-arrestin signaling is a promising strategy to obtain a new class of antinociceptives devoid of adverse and drug tolerance effects. TRV130 is the first such a candidate introduced into the phase III clinical trial for treatment of acute severe pain [66]. Hence, more detailed understanding of structural aspects of interactions between pharmacologically relevant receptor and their biased ligands gives novel therapeutic perspective.

## Figures and Tables

**Figure 1 molecules-26-00851-f001:**
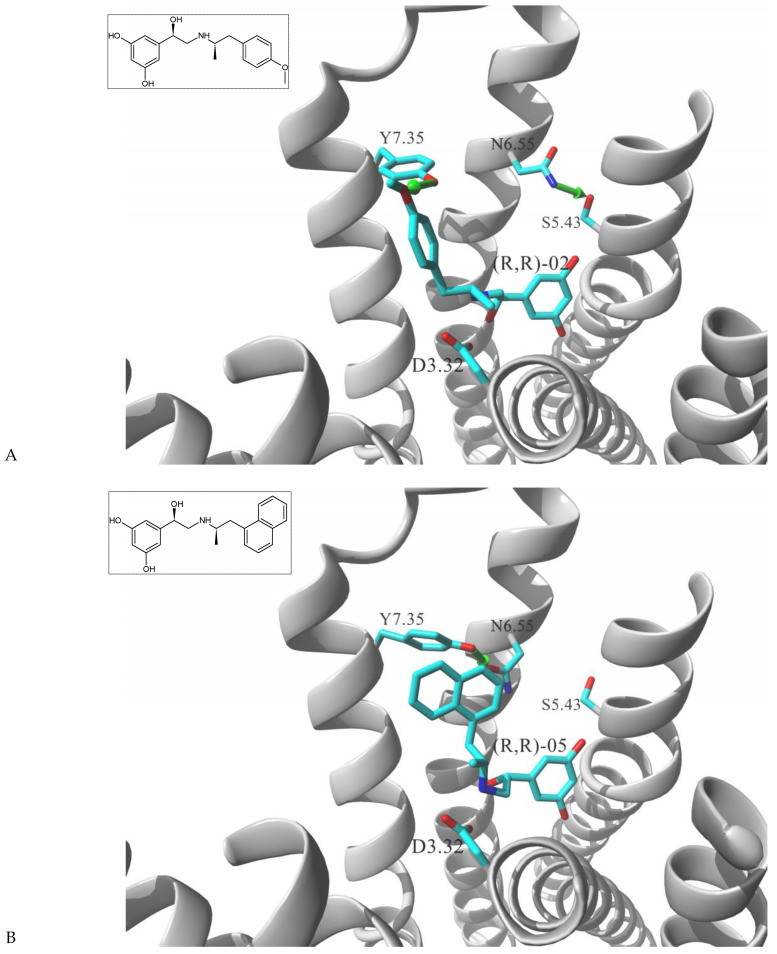
Postulated mechanism of differential interactions between β_2_-AR molecule and (**A**) (*R*,*R*)-02, (a ligand inducing G_s_ signaling pattern), or (**B**) (*R*,*R*)-05, a ligand inducing [G_s_+G_i_] signaling pattern. The receptor molecule is shown in secondary structure mode (gray ribbon), parts are hidden for clarity. Ligands molecules and Y308^7.35^, N293^6.55^, S204^5.43^ and D113^3.32^ residues are rendered in stick mode, key HBs discussed in the text are shown as green arrows. Inset: chemical formula of a ligand. Adopted from [27].

**Figure 2 molecules-26-00851-f002:**
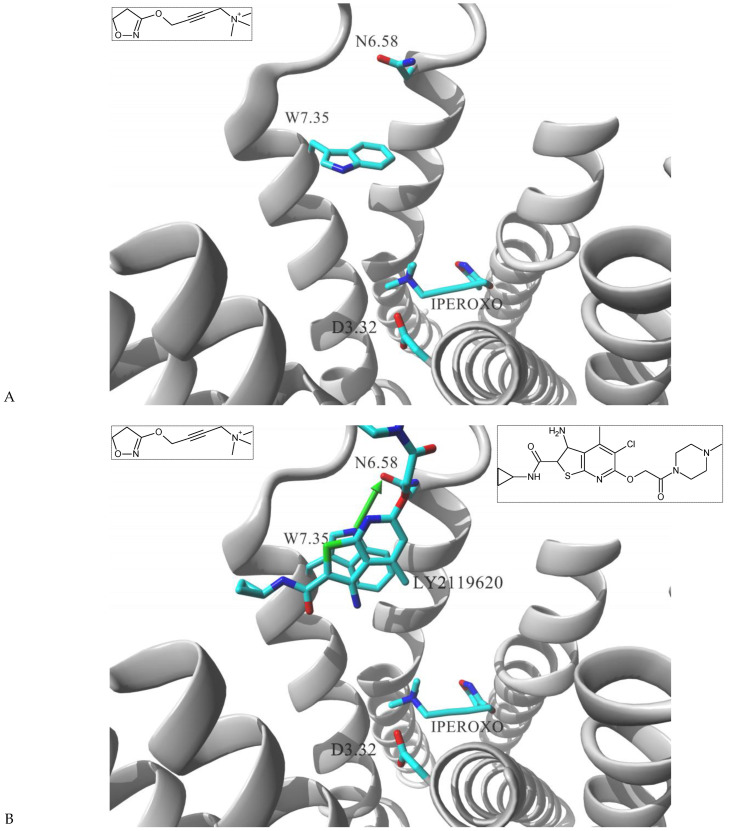
Structural model of M2 receptor cocrystalized with (**A**) iperoxo alone (4mqs.pdb) and (**B**) iperoxo + LY2119620 ligands (4mqt.pdb) [44]. Iperoxo resides at the ortosteric site, LY2119620 occupies the allosteric ELBD. The receptor molecule is shown in secondary structure mode (gray ribbon), parts are hidden for clarity. Ligand molecules and W422^7.35^, N410^6.58^ and D103^3.32^ residues are rendered in stick mode, interhelical HB in (**B**) is shown as a green arrow. Inset: chemical formula of ligands.

**Figure 3 molecules-26-00851-f003:**
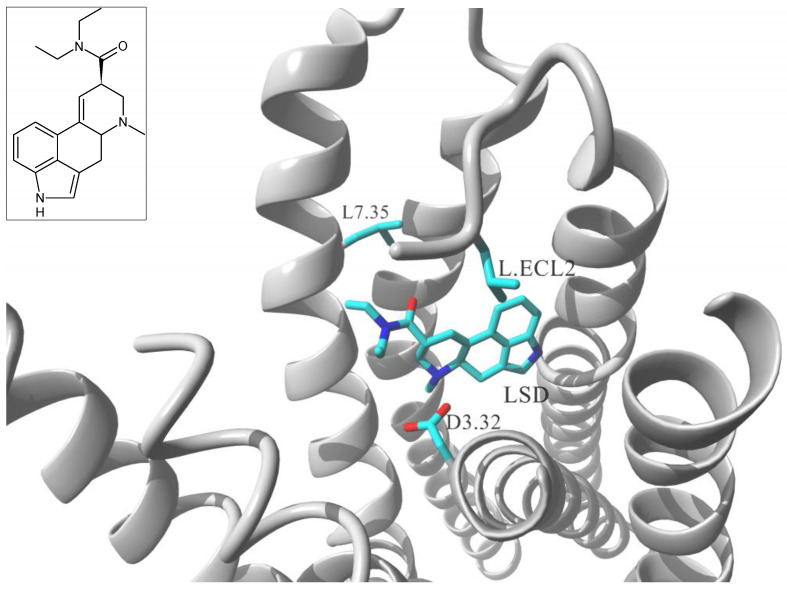
Structural model of 5-HT_2B_ receptor cocrystalized with LSD (5tvn.pdb) [45] and location of the two ELBD residues, L362^7.35^, L209^ECL2^, postulated to be involved in biased signaling. The receptor molecule is shown in secondary structure mode (gray ribbon), parts are hidden for clarity. Ligand molecules and L362^7.35^, L209^ECL2^ and D135^3.32^ residues are rendered in stick mode. Inset: chemical formula of a ligand.

**Figure 4 molecules-26-00851-f004:**
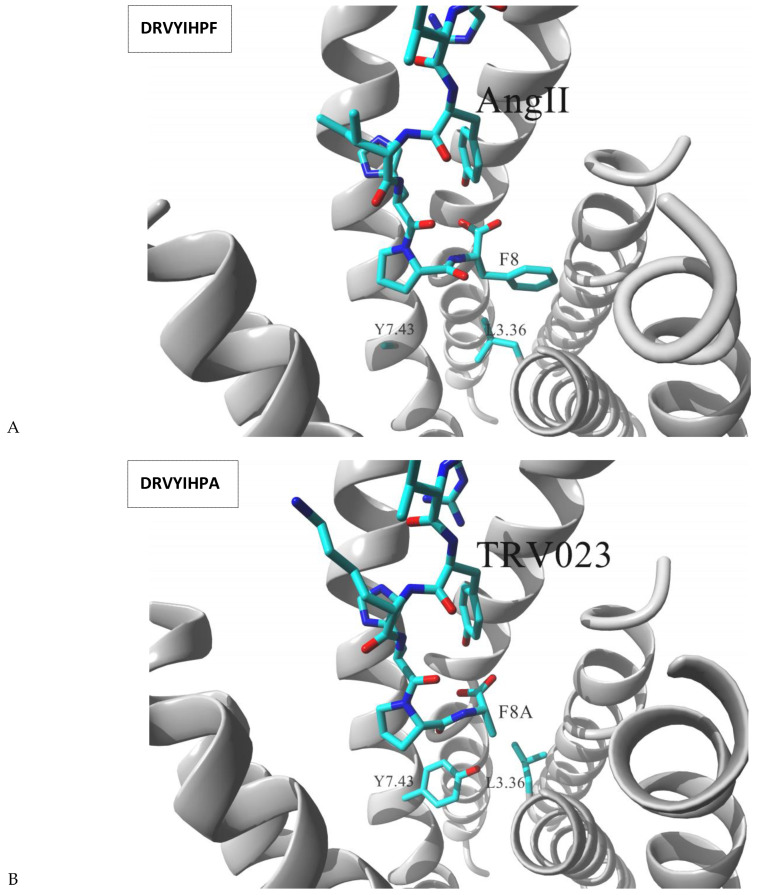
AT_1_ receptor cocrystalized with (**A**) wild type AngII peptide (6os0.pdb) (G_q_ + β-arrestin signaling) and (**B**) F8A AngII mutant (TRV023; 6os1.pdb) (β-arrestin biased signaling). Due to high B-factor, conformation of the Y292^7.43^ cannot be fully visualized in (**A**) [58]. The receptor is shown in secondary structure mode (gray ribbon), parts are hidden for clarity. Ligand molecules, L112^3.36^ and Y292^7.43^ residues are rendered in stick mode. Inset: sequence of cocrystalized peptide.

**Table 1 molecules-26-00851-t001:** Chemical structures of fenoterol derivatives inducing ligand directed signaling of β_2_-AR at rat cardiomyocytes, controlled by PTx [27].

G_s_ Signaling Ligands (PTx Insensitive)	[G_s_+G_i_] Signaling Ligands (PTx Sensitive)
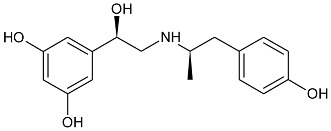 (*R*,*R*)-01	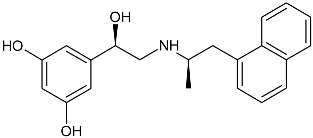 (*R*,*R*)-05
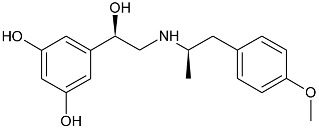 (*R*,*R*)-02	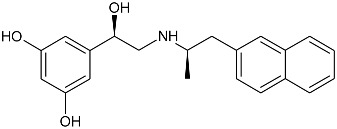 (*R*,*R*)-06
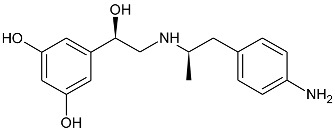 (*R*,*R*)-03	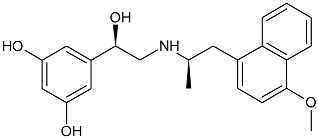 (*R*,*R*)-07

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
