# Peer review of "Structural Insights into Ligand—Receptor Interactions Involved in Biased Agonism of G-Protein Coupled Receptors"

_molecules, 2021, doi:10.3390/molecules26040851_

Round 1
Reviewer 1 Report
This is a modest and reasonably interesting review that covers some structural elements of the concept of ligand bias in GPCRs. This is a notable topic of current interest and the manuscript is likely to find a reasonable number of interested readers. The paper, as far as it goes, represents some valuable results from this topic in a generally accurate way that is easy for the reader to follow. However, it is very far from complete in that there are numerous examples in the opioid and dopamine areas of strong bias that are not included or even referenced in the review. This is fine, in principle, but if the intent of the authors is to provide a limited view of the field, they should at least note the scope of their intended coverage so that readers unfamiliar with the field are not left with the idea that this is all that there is to ligand bias. In addition, it would aid the reader if all chemical structures noted in the text or visualized in the three-dimensional models presented were explicitly drawn in standard forms.
Author Response
We appreciate the reviewer comments. Indeed, the work was not intended to be a complete review of the biased agonisms across different class A GPCRs, we wanted to describe interesting structural features in ligand - receptor interactions which seems to be common and shared across receptors. In revised manuscript the goal of the work was described more precisely, we also included a number of new citations which refer the readers to the articles dealing more completely with bias properties in specific receptor types. In addition, chemical formula of ligand molecules has been included and insets to each figure showing ligand - receptor complex.
We have decided to include one additional figure illustrating the role of ECL2 lid on the extended ligand binding domain within 5-HT2B receptor.
Reviewer 2 Report
The review by Krzysztof JóĹşwiak and Anita PĹ‚aziĹ„ska provides structural insights into GPCR-ligand interactions underlying biased signaling. Summarized are data for extensively studied and druggable class A GPCRs including β-AR, D2L, M2, 5HR2B, opioid, and AT1 receptors. This is a well-written up-to-date review.
Minor comments:
There are a few typos/grammatical errors and very long sentences that need to be corrected/rewritten (a few examples are given below).
Line 30 - "structural features of an individual GPCR allow precise recognition of defined extracellular" - should read "structural features of an individual GPCR allow precise recognition of defined extracellular".
Lines 54 - 60 - The sentence "Analyses of structure-activity relationships" is too long and difficult to understand. Please consider rewriting it.
Line 173 - "Tschammer et al. in his work" - should read " Tschammer et al. in their work".
Line 449 - "maintain relatively uniform molecular mechanisms conformational" - should probably read as: "maintain relatively uniform molecular mechanisms of conformational".
Lines 455 - 456 - "Combination of medicinal chemistry, molecular pharmacology, structural biology, mutagenesis studies" - should probably read as: "Combination of medicinal chemistry, molecular pharmacology, structural biology and mutagenesis studies".
Lines 496 - 3.36 - superscript.
Captions need to be placed below the figures.
Author Response
We appreciate the reviewer comments. The manuscript has been double checked for typos. We have paid additional attention to long and complex sentences. Captions have been moved below figures.